# Novel Hybrid Nanomaterials Based on Poly-*N*-Phenylanthranilic Acid and Magnetic Nanoparticles with Enhanced Saturation Magnetization

**DOI:** 10.3390/polym14142935

**Published:** 2022-07-20

**Authors:** Sveta Zhiraslanovna Ozkan, Aleksandr Ivanovich Kostev, Petr Aleksandrovich Chernavskii, Galina Petrovna Karpacheva

**Affiliations:** 1A.V. Topchiev Institute of Petrochemical Synthesis, Russian Academy of Sciences, 29 Leninsky Prospect, 119991 Moscow, Russia; kostev@ips.ac.ru (A.I.K.); chern5@inbox.ru (P.A.C.); gpk@ips.ac.ru (G.P.K.); 2Department of Chemistry, Lomonosov Moscow State University, 1–3 Leninskie Gory, 119991 Moscow, Russia

**Keywords:** magnetic nanomaterials, conjugated polymers, poly-*N*-phenylanthranilic acid, one-step synthesis, IR heating

## Abstract

A one-step preparation method for cobalt- and iron-containing nanomaterials based on poly-*N*-phenylanthranilic acid (P-N-PAA) and magnetic nanoparticles (MNP) was developed for the first time. To synthesize the MNP/P-N-PAA nanocomposites, the precursor is obtained by dissolving a Co (II) salt in a magnetic fluid based on Fe_3_O_4_/P-N-PAA with a core-shell structure. During IR heating of the precursor in an inert atmosphere at *T* = 700–800 °C, cobalt interacts with Fe_3_O_4_ reduction products, which results in the formation of a mixture of spherical Co-Fe, γ-Fe, β-Co and Fe_3_C nanoparticles of various sizes in the ranges of 20 < *d* < 50 nm and 120 < *d* < 400 nm. The phase composition of the MNP/P-N-PAA nanocomposites depends significantly on the cobalt concentration. The reduction of metals occurs due to the hydrogen released during the dehydrogenation of phenylenamine units of the polymer chain. The introduction of 10–30 wt% cobalt in the composition of nanocomposites leads to a significant increase in the saturation magnetization of MNP/P-N-PAA (*M_S_* = 81.58–149.67 emu/g) compared to neat Fe_3_O_4_/P-N-PAA (*M_S_* = 18.41–27.58 emu/g). The squareness constant of the hysteresis loop is *κ_S_* = *M_R_*/*M_S_* = 0.040–0.209. The electrical conductivity of the MNP/P-N-PAA nanomaterials does not depend much on frequency and reaches 1.2 × 10^−1^ S/cm. In the argon flow at 1000 °C, the residue is 77–88%.

## 1. Introduction

Nanocomposite magnetic materials based on polyconjugated polymers are new generation materials with physical and chemical properties required for modern technologies [1,2,3,4,5]. These magnetic nanocomposites can find potential application as hybrid electrocatalysts [6,7], as cathode materials for rechargeable batteries and fuel cells [8,9,10], as active materials in solar cells [11,12,13] and supercapacitors [14,15,16,17,18], for the remediation of water resources [19,20,21,22,23,24], ion-exchange materials [25,26,27], ion-specific electrodes [28,29,30,31], to produce sensors [32,33,34,35,36], as anticorrosive coatings [37,38,39,40], as electromagnetic radiation absorbing materials [41,42,43,44,45,46,47,48,49,50] and for medical applications (as antibacterial and antifungal agents, for controlled drug release) [38,51,52]. Moreover, high-magnetic-moment materials are basic components of devices used in electronic and optical fields. Wireless charging can be carried out by absorbing emitted from the power generators electromagnetic waves [53,54].

Progress in the creation of novel multifunctional magnetic nanomaterials is due to the development of new synthesis methods for nanostructures. The most common method of obtaining hybrid magnetic nanocomposites is oxidative polymerization of monomers (aniline [8,9,25,29,32,51,55,56], pyrrole [28,40,44,57] and thiophene [23]) in a reaction medium containing prefabricated magnetic nanoparticles (Fe_3_O_4_ [25,28,29,42,55,58], *γ*-Fe_2_O_3_ [32,43], *α*-Fe_2_O_3_ [37], Co_3_O_4_ [8,9,38,50] and CoFe_2_O_4_ [33,59]). One of the ways to prevent the metal nanoparticles aggregation is their stabilization in a polymer matrix during synthesis. Development of a complex synthetic strategy would make it possible to expand the range of magnetic nanoparticles based nanomaterials.

Hybrid magnetic nanomaterials with core-shell structure, where Fe_3_O_4_ nanoparticles form the core and poly-*N*-phenylanthranilic acid (P-N-PAA) is the shell, was obtained by a one-pot method [60]. The originality of the one-pot synthesis method lies in the fact that the Fe_3_O_4_/P-N-PAA preparation does not require prefabricated Fe_3_O_4_ nanoparticles. At the same time, the entire process of nanocomposites synthesis without intermediate stages of product extraction and purification is carried out in the same reaction vessel. The obtained nanomaterials are superparamagnetic. However, the saturation magnetization *M_S_* does not exceed 27.58 emu/g. 

Inverse suspension polymerization was applied to cover iron-magnetic materials with conjugated polymers. Prepared from inverse suspension-polymerized Fe_3_O_4_/polyaniline composites were subjected to calcination at 950 °C in the argon atmosphere to synthesize α-Fe (ferrite), Fe_3_C (cementite) and α^//^-Fe_x_N_y_ (ferric nitride) based high-magnetic-moment materials [53,54].

Earlier, we have proposed an original preparation method of magnetic nanomaterials under IR heating of polydiphenylamine (PDPA) and polyphenoxazine (PPOA) in the presence of metal salts [61,62,63,64,65]. It was shown that the phase state, as well as size and shape of the resulting magnetic particles depend on the polymer matrix nature. At the same time, an important role belongs to the polymer thermal stability and the presence of a sufficient amount of hydrogen in its structure capable of reducing metals during dehydrogenation. Under the same conditions of synthesis, IR heating of PDPA in the presence of Co(OOCCH_3_)_2_·4H_2_O leads to the formation of α-Co and β-Co nanoparticles, their size ranging 2 < *d* < 8 nm. In the case of PPOA, only β-Co nanoparticles are formed, their size ranging 4 < *d* < 14 nm. IR heating of the precursor obtained by co-solution of PDPA or PPOA and salts of cobalt Co (II) and iron Fe (III) in an organic solvent leads to the formation only of bimetallic Co-Fe particles dispersed in a polymer matrix. For Co-Fe/PDPA and Co-Fe/PPOA nanomaterials, the saturation magnetization *M_S_* is not higher than 20.43 and 27.28 emu/g, respectively (Table 1). Thus, magnetic nanomaterials with saturation magnetization exceeding *M_S_* = 20–27 emu/g cannot be obtained by the proposed methods. 

The development of novel high-magnetic-moment materials seems to be an urgent task. The aim of the proposed work is to create nanocomposite magnetic materials with high saturation magnetization. 

In this work, a one-step preparation method for cobalt- and iron-containing nanomaterials based on poly-*N*-phenylanthranilic acid (P-N-PAA) and magnetic nanoparticles (MNP) with enhanced saturation magnetization was developed for the first time. The nanocomposites were obtained by IR heating of precursors prepared by dissolving a Co (II) salt in a magnetic fluid based on Fe_3_O_4_/P-N-PAA with a core-shell structure. An analysis of the phase composition, morphology, magnetic, electrical and thermal properties of MNP/P-N-PAA nanomaterials depending on the conditions of synthesis was carried out. 

## 2. Experimental Section

### 2.1. Materials

N-phenylanthranilic acid (diphenylamine-2-carboxylic acid) (C_13_H_11_O_2_N) (analytical grade), chloroform (reagent grade), aqueous ammonia (reagent grade), sulfuric acid (reagent grade), DMF (Acros Organics, 99%), as well as iron (II) sulfate (Acros organics), iron (III) chloride (high purity grade) and Co(OOCCH_3_)_2_·4H_2_O (pure grade), were used without further purification. Ammonium persulfate (analytical grade) was purified by recrystallization from distilled water. Aqueous solutions of reagents were prepared using distilled water. 

### 2.2. Synthesis of Fe_3_O_4_/P-N-PAA

Fe_3_O_4_/P-N-PAA was synthesized by a method of one-pot synthesis in an interfacial process developed by the authors in [60]. To obtain the Fe_3_O_4_/P-N-PAA nanocomposite, firstly, Fe_3_O_4_ nanoparticles of the required concentration were synthesized via hydrolysis of a mixture of iron (II) and (III) salts with a molar ratio of 1:2 in an ammonium hydroxide solution at 55 °C. For that, 0.43 g of FeSO_4_⋅7H_2_O and 1.175 g of FeCl_3_⋅6H_2_O were dissolved in 20 mL of distilled water and heated to 55 °C, then 5 mL of NH_4_OH were added. A N-PAA solution (0.1 mol/L, 1.0 g) in a mixture of chloroform (60 mL) and NH_4_OH (5 mL) (volume ratio is 12:1) was added to the obtained aqueous alkaline suspension of Fe_3_O_4_ nanoparticles. The process was carried out at 55 °C under constant intensive stirring for 0.5 h. The suspension was cooled at room temperature under constant intensive stirring for 1 h. Then, for the in situ interfacial oxidative polymerization of N-PAA on the surface of Fe_3_O_4_, an aqueous solution (0.2 mol/L, 1.96 g) of ammonium persulfate (30 mL) was added to the Fe_3_O_4_/N-PAA suspension thermostated under constant stirring at 0 °C. Solutions of the organic and aqueous phases were mixed immediately without gradual dosing of reagents. The volume ratio of the organic and aqueous phases is 1:1 (V_total_ = 120 mL). The polymerization reaction continued for 3 h under constant intensive stirring at 0 °C. When the synthesis was completed, the reaction mixture was precipitated in a threefold excess of 1 M H_2_SO_4_. The obtained product was filtered off, washed repeatedly with distilled water until neutral reaction, and then vacuum-dried over KOH to constant weight. The Fe_3_O_4_/P-N-PAA yield is 0.741 g. The content of iron in the resulting Fe_3_O_4_/P-N-PAA is [Fe] = 16.4% (according to ICP-AES data). 

The Fe_3_O_4_/P-N-PAA nanocomposite suspension in DMFA was prepared to obtain magnetic fluids. The stability of suspension was being observed for 6 months.

### 2.3. Synthesis of MNP/P-N-PAA

The following method was used to obtain the MNP/P-N-PAA nanocomposite [66]. To prepare a magnetic fluid, 0.2 g of Fe_3_O_4_/P-N-PAA were added to 15 mL of DMF in a 100 mL crystallization dish. Then, Co(OOCCH_3_)_2_·4H_2_O of required concentration was dissolved in the resulting stable suspension. The Fe_3_O_4_/P-N-PAA concentration in the DMF solution was 2 wt%. In the initial Fe_3_O_4_/P-N-PAA nanocomposite, the content of iron is [Fe] = 16.4 and 38.5% (according to ICP-AES data). The content of cobalt is [Co] = 5–30 wt% relative to the weight of the neat nanocomposite. After removing the solvent (DMF) at *T* = 85 °C, the precursor consisting of Fe_3_O_4_/P-N-PAA and cobalt acetate salt was subjected to IR radiation using an automated IR heating unit [61] in an argon atmosphere at *T* = 700–800 °C for 2 min. The heating rate was 50 °C min^−1^. The yield of MNP/P-N-PAA is 0.109 g (51.39%) at the cobalt content [Co] = 30 wt% relative to the Fe_3_O_4_/P-N-PAA weight. Figure 1 shows a flowchart with the step-by-step preparation of MNP/P-N-PAA nanomaterials.

### 2.4. Characterization

The IR heating unit, which is a laboratory quartz tube IR furnace [61], was used to synthesize the MNP/P-N-PAA nanocomposites. The halogen lamps were implemented as a radiation source. The rectangular graphite box with the samples was placed in a quartz reactor. The heating temperature was regulated via IR radiation intensity.

An inductively coupled plasma atomic emission spectroscopy method (ICP-AES) was used to measure quantitatively the metal content in the nanocomposites on a Shimadzu ICP emission spectrometer (ICPE-9000) (Kyoto, Japan).

A Difray-401 X-ray diffractometer (Scientific Instruments Joint Stock Company, Saint-Petersburg, Russia) with Bragg–Brentano focusing on Cr*K*_α_ radiation, *λ* = 0.229 nm was used to perform in air X-ray diffraction study.

A Bruker IFS 66v FTIR spectrometer (Karlsruhe, Germany) was used to measure FTIR spectra in the range of 400–4000 cm^−1^. The samples were prepared as KBr pressed pellets.

A Senterra II Raman spectrometer (Bruker, Karlsruhe, Germany) was used to record Raman spectra using a laser with the wavelength of 532 nm and the power of 0.25 mW, spectral resolution of 4 cm^−1^. 

A JEM-2100 transmission electron microscope (accelerating voltage of 200 kV) (JEOL, Akishima, Tokyo, Japan) and a Hitachi TM 3030 scanning electron microscope (Hitachi High-Technologies Corporation, Fukuoka, Japan) with magnification up to 30,000 and 30 nm resolution were used to perform an electron microscopic study. The size of nanoparticles is determined using the EsiVision software.

A vibration magnetometer was used to measure specific magnetization at room temperature depending on the magnetic field value [67].

An E7-20 precision LCR-meter (MC Meratest, Moscow, Russia) was used to measure the *ac* conductivity in the frequency range of 25.0 Hz–1.0 MHz. To measure the frequency dependence on the conductivity (σ_*ac*_), samples as a tablet with a diameter of 6 mm and a thickness of 3–5 mm, pressed into a mold made of PTFE, were prepared. Brass electrodes were located on both sides of the mold. Tablet compression was performed by pressing the powder in the PTFE mold with a threaded connection. The design of the measuring cell was similar to the “swagelok cell”, but without the spring. The torque force was 10 N⋅m. 

A Mettler Toledo TGA/DSC1 (Giessen, Germany) was used to perform thermogravimetric analysis (TGA) in the dynamic mode in the range of 30–1000 °C in air and in the argon flow. The heating rate was 10 °C/min, and the argon flow velocity was 10 mL/min. The samples were analyzed in an Al_2_O_3_ crucible.

A Mettler Toledo DSC823^e^ calorimeter (Giessen, Germany) was used to perform differential scanning calorimetry (DSC). The heating rate was 10 °C/min in the nitrogen atmosphere, with the nitrogen flow rate of 70 mL/min. 

## 3. Results and Discussion

### 3.1. Characterization of Nanomaterials

A one-step method was proposed to synthesize novel hybrid nanomaterials based on P-N-PAA and cobalt- and iron-containing magnetic nanoparticles (MNP). The choice of polymer was due to the fact that, unlike PDPA, the presence of carboxyl groups in the P-N-PAA structure promotes the formation of a nanomaterial with a core-shell structure, where Fe_3_O_4_ nanoparticles form the core, and P-N-PAA is the shell. The polymer shell effectively prevents the aggregation of nanoparticles, which makes it possible to use Fe_3_O_4_/P-N-PAA nanocomposites to obtain magnetic fluids suitable for dissolving metal salt. When the precursor obtained from Fe_3_O_4_/P-N-PAA and a Co (II) salt is IR heated in an inert atmosphere at *T* = 700–800 °C, metals are reduced due to the hydrogen released during dehydrogenation of phenylenamine units with the formation of a mixture of Co-Fe, γ-Fe, β-Co and Fe_3_C magnetic nanoparticles. As for the core-shell structure, due to the reduction of Fe_3_O_4_, a Fe---OOC coordination bond formed via binding of the carboxylate-ion with iron in Fe_3_O_4_/P-N-PAA is broken. As a result, MNP/P-N-PAA nanocomposite materials are formed. They are cobalt- and iron-containing MNP of various compositions dispersed in a polyconjugated polymer matrix. Figure 2 shows a synthesis scheme of the MNP/P-N-PAA nanomaterials. 

The originality and distinctive feature of the proposed approach to the MNP/P-N-PAA synthesis is determined by the fact that the precursor is obtained via dissolving a Co (II) salt in a magnetic fluid based on the Fe_3_O_4_/P-N-PAA nanocomposite, which was synthesized by the authors and which has a core-shell structure [60]. In DMF, Fe_3_O_4_/P-N-PAA forms magnetic fluids combining the properties of a magnetic material and a liquid. That is, the presence of a polymer coating prevents the aggregation of Fe_3_O_4_ nanoparticles, ensuring the magnetic fluid stability over 6 months. An important role belongs to the hydrogen amount in the structure of polymer capable of reducing metals during dehydrogenation. During the synthesis of MNP/P-N-PAA from Fe_3_O_4_/P-N-PAA, cobalt interacts with Fe_3_O_4_ reduction products, which leads to the formation of cobalt- and iron-containing MNP of different compositions. Whereas, IR heating of the precursor obtained by co-solution of the polymer and salts of cobalt Co (II) and iron Fe (III) in an organic solvent leads to the formation only of bimetallic Co-Fe particles dispersed in a polymer matrix [61].

The novelty of the proposed approach is determined by the fact that the use of Fe_3_O_4_/P-N-PAA based magnetic fluid for dissolving the Co (II) salt makes it possible to expand the range of high-magnetic-moment nanoparticles in the nanomaterial composition. The developed one-step method for the formation of a nanocomposite material under IR heating helps to obtain Co-Fe, γ-Fe, β-Co and Fe_3_C nanoparticles of various compositions directly during the synthesis of the nanocomposite without subjecting the polymer matrix to destruction. At the same time, applying incoherent IR radiation in a pulsed mode for the formation of the magnetic nanomaterial can significantly reduce energy costs. 

As shown in Figure 3, IR heating of Fe_3_O_4_/P-N-PAA at 600 °C does not lead to any noticeable phase changes. XRD patterns of Fe_3_O_4_/P-N-PAA and Fe_3_O_4_/P-N-PAA, IR heated at 600 °C, demonstrate only the Fe_3_O_4_ phase, clearly identified by its reflection peaks in the range of scattering angles 2θ = 46.3°, 54.6°, 66.8°, 84.7°, 91.0° and 101.6° (Cr*K*_α_ radiation) [60]. These diffraction peaks correspond to Miller indices (220), (311), (400), (422), (511) and (440) and characterize the simple cubic lattice of Fe_3_O_4_ (JCPDS 19-0629) [68]. At 800 °C, partial reduction of Fe_3_O_4_ with the formation of FeO, α-Fe, γ-Fe and Fe_4_N nanoparticles takes place. 

IR heating of Fe_3_O_4_/P-N-PAA in the presence of Co (II) salt at 800 °C leads to the formation of a mixture of Co-Fe, γ-Fe, β-Co and Fe_3_C magnetic nanoparticles, which was confirmed by X-ray phase analysis (Figure 4). In this case, the phase composition of the MNP/P-N-PAA nanocomposites depends on the cobalt concentration. In the FTIR spectrum of MNP/P-N-PAA, the absence of an intense absorption band at 572 cm^−1^, corresponding to the stretching vibrations of the ν_Fe–O_ bond, is explained by Fe_3_O_4_ reduction (Figure 5). 

According to elemental analysis data, when Fe_3_O_4_/P-N-PAA is IR heated in the presence of Co(CH_3_CO_2_)_2_·4H_2_O, the dehydrogenation of phenylenamine units causes the decrease in the content of hydrogen from 1.9 to 0.1% (the C/H ratio increases from 12.5 to 70.7) (Table 2). The released hydrogen contributes to the reduction of metals. The C/N ratio changes insignificantly (from 8.6 to 9.0), which indicates that the polymer component degradation during IR heating is weakly expressed.

Figure 6 shows Raman spectra of P-N-PAA, neat Fe_3_O_4_/P-N-PAA and Fe_3_O_4_/P-N-PAA, IR heated at 600 and 800 °C, and MNP/P-N-PAA. As seen in Figure 6, as well as in the IR heated at 800 °C Fe_3_O_4_/P-N-PAA, there are two pronounced bands: a G band at 1596 cm^−1^ from sp^2^ (aromatic) carbon atoms and a D band at 1350 cm^−1^ from sp^3^ carbon atoms in the Raman spectrum of MNP/P-N-PAA. In the Raman spectra of both the neat Fe_3_O_4_/P-N-PAA and Fe_3_O_4_/P-N-PAA, IR heated at 600 °C, as well as in P-N-PAA, these bands are absent. The intensity ratio of these bands (I_D_/I_G_ = 0.82), the high intensity and the width of the 2D band at 2800 cm^−1^ indicate that graphite-like structures are formed in the polymer component structure during IR heating of the precursor based on Fe_3_O_4_/P-N-PAA and a Co (II) salt in an inert atmosphere at 800 °C. XRD patterns of MNP/P-N-PAA nanocomposites demonstrate a wide halo in the range of scattering angles 2θ = 20–44°, which characterizes graphite-like structures (Figure 4). The broadening of all main absorption bands characterizing the chemical structure of the polymer component in the FTIR spectrum of MNP/P-N-PAA is also connected with the dehydrogenation of P-N-PAA phenylenamine units and the subsequent formation of graphite-like structures (Figure 5). 

However, it should be noted that the presence of nitrogen and hydrogen atoms in the MNP/P-N-PAA structure (Table 2), as well as the absence of a sharp peak at diffraction angles 2θ = 39.36°, which characterizes carbon, on the XRD patterns of MNP/P-N-PAA (Figure 4) indicate an incomplete transformation of the polymer component to all-carbon structures. Apparently, during the reduction of Fe_3_O_4_ in the course of Fe_3_O_4_/P-N-PAA thermal transformations at high temperatures (700–800 °C), the partial formation of graphite-like structures occurs. 

XRD patterns of all MNP/P-N-PAA nanocomposites show clearly identified reflection peaks of Co-Fe nanoparticles in the region of diffraction scattering angles 2θ = 68.86°, 106.36° (Figure 4). According to the database Miller indices, interplanar distances in the Co-Fe phase correspond to the Co-Fe solid solution. The formation of bimetallic Co-Fe nanoparticles is associated with the interaction of cobalt and Fe_3_O_4_ reduction products. The reflection peaks of β-Co nanoparticles with a face-centered cubic lattice can be identified at diffraction angles 2θ = 67.52°, 80.14°. In the case of γ-Fe nanoparticles they are identified in the region of 2θ = 66.56°, 78.77°. The formation of Fe_3_C nanoparticles at [Co] = 5 wt% is associated with the interaction of Fe_3_O_4_ reduction products with the polymer matrix due to low concentration of cobalt.

According to TEM and SEM data, a bimodal nature of the MNP distribution is observed. A mixture of spherical nanoparticles sized 20 < *d* < 50 nm and 120 < *d* < 400 nm is formed (Figure 7 and Figure 8). At the same time, according to XRD data for the MNP/P-N-PAA nanocomposite, the CSR size distribution curve is in the region of 3–55 nm with a peak at 13–14 nm (Figure 9). According to ICP-AES data, depending on the conditions of synthesis the content of Co = 8.6–38.0%, and Fe = 14.2–58.3% (Table 2 and Table 3). According to elemental analysis data, the content of P-N-PAA component is 12.6–72.2%. 

Energy dispersive X-ray spectroscopy (EDS) elemental mapping method was used to characterize the element distribution in the P-N-PAA based nanomaterials (Figure 10). Figure 11 demonstrates SEM-EDS mapping images of ferrum Fe, cobalt Co, carbon C, nitrogen N and oxygen O in Fe_3_O_4_/P-N-PAA and MNP/P-N-PAA. Table 4 shows the EDS analysis data of these nanocomposites. The EDS element mapping reveals a homogeneous distribution of Fe and Co elements. As seen in Table 4, the oxygen content in the MNP/P-N-PAA nanocomposite drops sharply due to the Fe_3_O_4_ reduction.

### 3.2. Magnetic Properties of Nanomaterials

The obtained MNP/P-N-PAA nanomaterials demonstrate a hysteresis character of remagnetization at room temperature. The dependency of magnetization on the value of the applied magnetic field is shown in Figure 12 and Figure 13. The residual magnetization *M_R_* of the MNP/P-N-PAA nanomaterials is 3–20 emu/g, the coercive force is *H_C_* = 45–200 Oe (Table 3). 

The squareness constant of the hysteresis loop is *κ_S_* = *M_R_*/*M_S_* = 0.040–0.209, which indicates a significant proportion of superparamagnetic nanoparticles. However, the content of superparamagnetic nanoparticles can be quantified only if the single-domain condition is satisfied. In this case, it is difficult to determine the quantity of superparamagnetic nanoparticles, since MNP are a mixture of spherical Co-Fe, γ-Fe, β-Co and Fe_3_C nanoparticles of various sizes in the ranges of 20 < *d* < 50 nm and 120 < *d* < 400 nm.

As shown in Figure 12, IR heating of neat Fe_3_O_4_/P-N-PAA at 600 °C has little effect on its magnetic properties (*M_S_* = 18.41–17.02 emu/g). At 800 °C, the saturation magnetization of Fe_3_O_4_/P-N-PAA decreases to 12.41 emu/g due to phase transformations of Fe_3_O_4_ into FeO, α-Fe, γ-Fe and Fe_4_N.

It can be seen from Table 3, that the introduction of 10–30 wt% cobalt into nanocomposites leads to a significant increase in the saturation magnetization of MNP/P-N-PAA compared to the initial Fe_3_O_4_/P-N-PAA. The saturation magnetization of MNP/P-N-PAA nanomaterials grow with the increase in cobalt concentration and reaches *M_S_* = 81.58–99.86 emu/g. An increase in the content of Fe_3_O_4_ nanoparticles in the initial Fe_3_O_4_/P-N-PAA nanocomposite ([Fe] = from 16.4 to 38.5%) leads to an even greater increase in the saturation magnetization of MNP/P-N-PAA—up to 149.67 emu/g, while in Fe_3_O_4_/P-N-PAA the value of *M_S_* is 27.58 emu/g (Figure 13). Furthermore, as shown in Table 1, IR heating of the precursor obtained by co-solution of the polymer and salts of cobalt Co (II) and iron Fe (III) in an organic solvent leads to the formation of Co-Fe-based magnetic nanomaterials with saturation magnetization not exceeding *M_S_* = 20–27 emu/g. Thus, the saturation magnetization of MNP/P-N-PAA grows with the increase in magnetic phase, but not linearly, since the MNP composition strongly depends on the Co and Fe content. 

### 3.3. Electrical Properties of Nanomaterials

The frequency dependence on *ac* conductivity of the MNP/P-N-PAA nanocomposite obtained at 800 °C during 2 min at [Co] = 30 wt% at the loading compared to neat Fe_3_O_4_/P-N-PAA was studied (Figure 14). The metallic phase composition corresponds to the Co-Fe and β-Co phases (Figure 4d).

The dependence of conductivity (σ_*ac*_) on the frequency is described by equation [69,70,71]:σ*_ac_* = σ*_*dc*_* _+_ *Aω*^n^

As can be seen from Table 5, neat P-N-PAA demonstrates a low conductivity value in the range of 25 Hz–1 MHz. The polymer electrical conductivity σ_*ac*_ increases linearly from 8.8 × 10^−11^ S/cm to 1.1 × 10^−7^ S/cm. The value of n = 0.75 indicates the hopping mechanism of conductivity (0 ≤ n ≤ 1), typical of most conductive polymers [69,70,71,72].

At low frequencies, the Fe_3_O_4_/P-N-PAA nanocomposite is characterized by weak dependence of electrical conductivity on frequency. As frequency grows, the Fe_3_O_4_/P-N-PAA electrical conductivity increases gradually by four orders of magnitude to 6.7 × 10^−6^ S/cm.

The electrical conductivity of the MNP/P-N-PAA nanomaterial is significantly higher than the conductivity of Fe_3_O_4_/P-N-PAA and does not depend much on frequency (1.1 × 10^−1^ S/cm–1.2 × 10^−1^ S/cm). The formation of an extended conjugated system during heat treatment of Fe_3_O_4_/P-N-PAA, as well as the presence of MNP of various compositions in the polymer matrix causes an increase in the degree of percolation and leads to a rise in electrical conductivity of MNP/P-N-PAA. Apparently, the formation of large conducting regions leads to exceeding the percolation threshold. 

As can be seen in Table 5, as well as for neat Fe_3_O_4_/P-N-PAA, for MNP/P-N-PAA the exponential parameter n lies in the range of 0 ≤ n ≤ 1, which is typical for systems with a hopping mechanism of charge transfer. The effect of tunneling in the nanocomposites is minimal. The *dc* conductivity plays an important role at low frequencies, whereas the *ac* conductivity of nanocomposites shows an increase with the growth in current frequency. 

### 3.4. Thermal Properties of Nanomaterials

Thermal stability of the MNP/P-N-PAA nanocomposites prepared at 800 °C for 2 min at [Co] = 5 and 30 wt% at the loading was studied by TGA and DSC methods. Figure 15 shows the dependence of temperature on the decrease in the weight of MNP/P-N-PAA compared to neat Fe_3_O_4_/P-N-PAA at heating up to 1000 °C in the argon flow and in air. 

The MNP/P-N-PAA nanocomposites are characterized by high thermal stability that exceeds considerably the thermal stability of neat Fe_3_O_4_/P-N-PAA. Weight loss at low temperatures on the TGA thermograms is associated with the removal of moisture (Figure 15). The DSC thermograms of nanomaterials show an endothermic peak at 108 °C (Figure 16). When re-heated, this peak is absent, which confirms the moisture removal. As can be seen, in air, after removing moisture, the same shape of weight loss curves until 500 °C is observed for all materials due to the thermooxidative degradation of polymer component. 

In an inert medium in MNP/P-N-PAA, a gradual weight loss is observed, and depending on the concentration of cobalt at 1000 °C, the residue is 77–88% (Table 6). At the same time, in Fe_3_O_4_/P-N-PAA, the residue accounts for 58%. In air, the lower thermal stability of MNP/P-N-PAA obtained at [Co] = 5 wt% is associated with the phase state of the MNP nanoparticles, as well as with a Fe_3_O_4_ content in neat Fe_3_O_4_/P-N-PAA ([Fe] = 16.4 wt% according to ICP-AES data) (Table 6). As can be seen on the TGA thermograms of MNP/P-N-PAA nanocomposites prepared at [Co] = 5 and 30 wt%, in air the content of the polymer component is 48 and 16%, respectively. The processes of thermooxidative degradation of MNP/P-N-PAA begin at 330 и 390 °C. The DTG curve of MNP/P-N-PAA shows removal of water and thermal decomposition of polymer component. The degradation of polymer component occurs within the range of 320–620 °C, with the maximum at 509 °C (Figure 17). Phase transformations of MNP nanoparticles occur in the range of 720–810 °C, with the maximum at 790 °C. 

## 4. Conclusions

The proposed one-step method offers the possibility of obtaining cobalt- and iron-containing nanomaterials based on the Fe_3_O_4_/P-N-PAA nanocomposite synthesized by the authors. Under IR heating of a precursor consisting of Fe_3_O_4_/P-N-PAA and a Co (II) salt, cobalt interacts with Fe_3_O_4_ reduction products to form a mixture of spherical Co-Fe, γ-Fe, β-Co and Fe_3_C nanoparticles with sizes of 20 < d < 50 nm and 120 < d < 400 nm. The phase composition of MNP/P-N-PAA nanocomposites depends on the cobalt concentration. The formation of the nanoparticles occurs directly during the nanocomposites synthesis under IR heating conditions, which makes it possible to expand the magnetic nanoparticles range. The originality of the proposed approach to MNP/P-N-PAA synthesis is determined by the fact that the preparation of a precursor by dissolving a Co (II) salt in a magnetic fluid based on Fe_3_O_4_/P-N-PAA with a core-shell structure leads to the formation of cobalt- and iron-containing nanomaterials with enhanced saturation magnetization. The introduction of 10–30 wt% cobalt into the composition of nanocomposites leads to a significant increase in the saturation magnetization of MNP/P-N-PAA (*M_S_* = 81.58–149.67 emu/g) compared to neat Fe_3_O_4_/P-N-PAA (*M_S_* = 18.41–27.58 emu/g). The hysteresis loop squareness constant *κ_S_* = *M_R_*/*M_S_* = 0.040–0.209 proves a significant share of superparamagnetic nanoparticles in MNP/P-N-PAA. The electrical conductivity of the MNP/P-N-PAA nanomaterials (1.1 × 10^−1^ S/cm–1.2 × 10^−1^ S/cm) is significantly higher than the conductivity of Fe_3_O_4_/P-N-PAA (6.7 × 10^−6^ S/cm) and does not depend much on frequency in the range of 25 Hz–1 MHz. The MNP/P-N-PAA nanocomposites are characterized by high thermal stability. In an inert atmosphere at 1000 °C, the residue is up to 88%, whereas the Fe_3_O_4_/P-N-PAA residue is 58%. The obtained magnetic nanomaterials can be used for modern technologies as materials that absorb electromagnetic radiation, to create wireless fast charging power source chargers, supercapacitors, electrochemical current sources, energy converters, contrast materials for magnetic resonance imaging, electromagnetic screens, in magnetic information recording systems, in high-temperature processes as protective coatings for construction materials, etc.

## Figures and Tables

**Figure 1 polymers-14-02935-f001:**
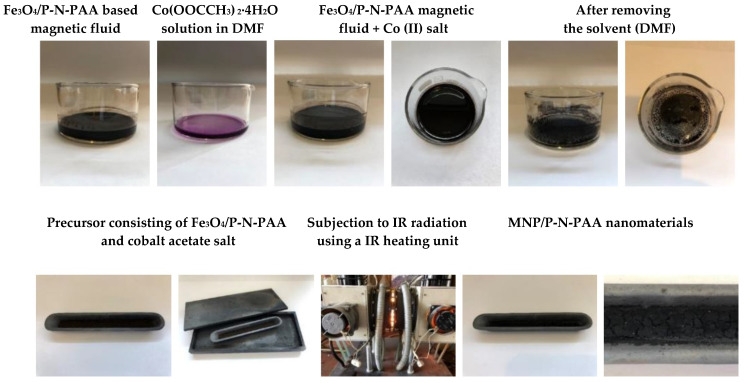
Flowchart with the step-by-step preparation of MNP/P-N-PAA nanomaterials.

**Figure 2 polymers-14-02935-f002:**
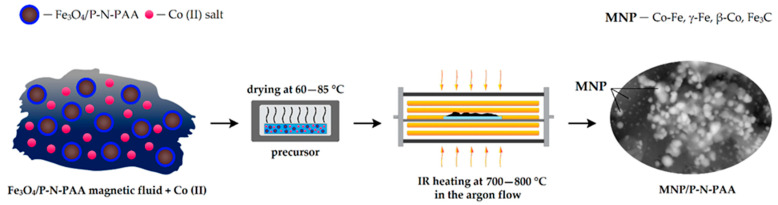
Scheme of the MNP/P-N-PAA nanomaterials synthesis.

**Figure 3 polymers-14-02935-f003:**
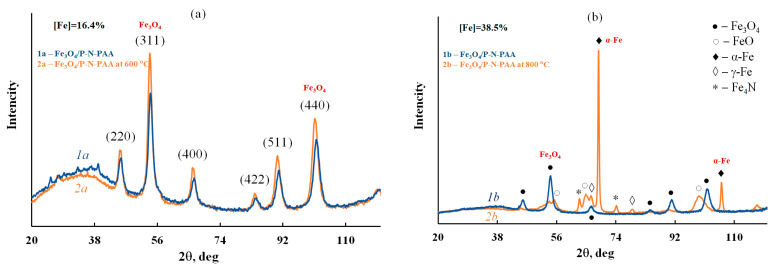
XRD patterns of Fe_3_O_4_/P-N-PAA (**1a**,**1b**) and Fe_3_O_4_/P-N-PAA, IR heated at 600 °C (**2a**) and 800 °C (**2b**).

**Figure 4 polymers-14-02935-f004:**
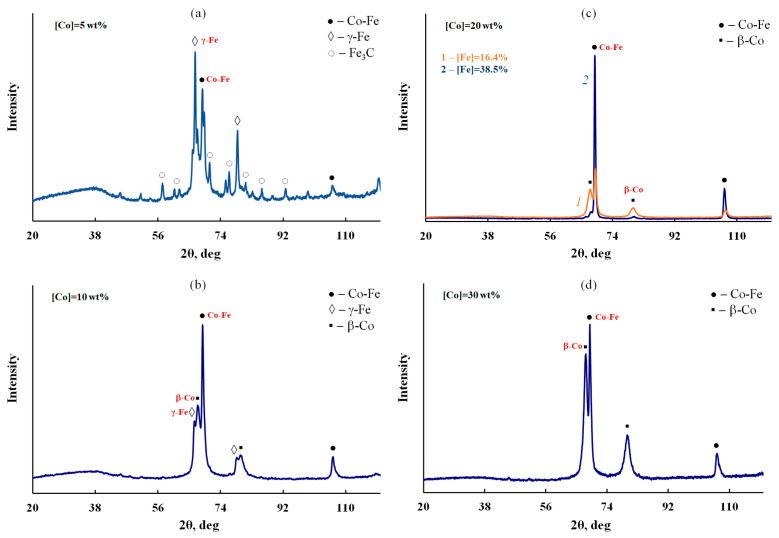
XRD patterns of MNP/P-N-PAA, obtained at 800 °C, [Co] = 5 (**a**), 10 (**b**), 20 (**c**) and 30 wt% (**d**).

**Figure 5 polymers-14-02935-f005:**
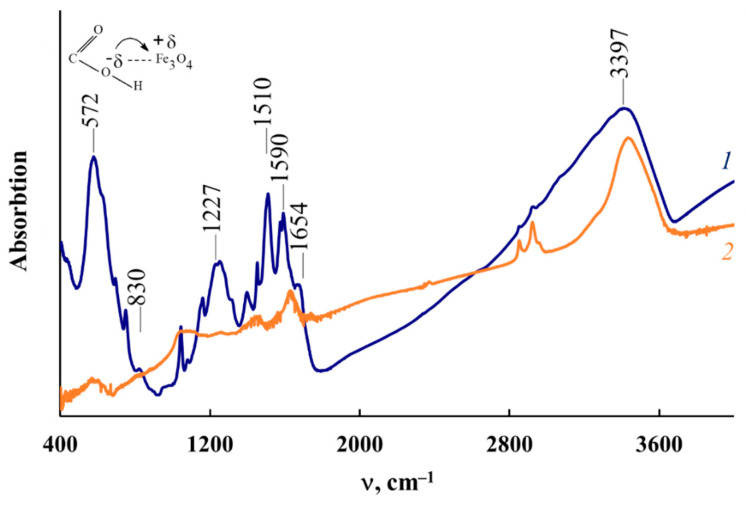
FTIR spectra of Fe_3_O_4_/P-N-PAA (*1*) and MNP/P-N-PAA (*2*).

**Figure 6 polymers-14-02935-f006:**
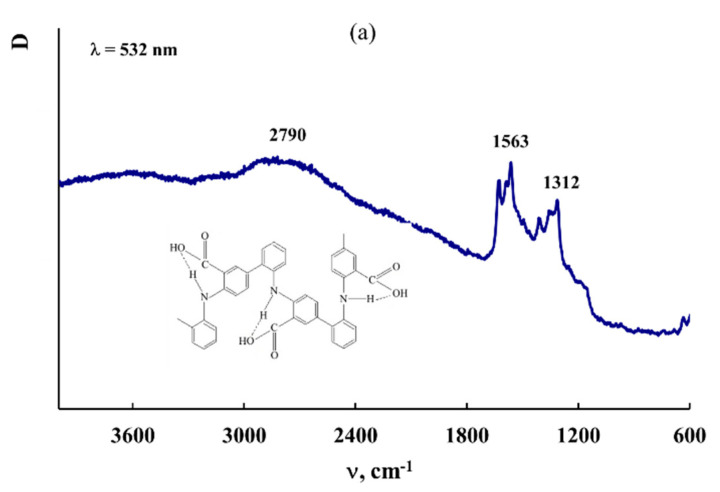
Raman spectra of P-N-PAA (**a**), Fe_3_O_4_/P-N-PAA (**1b**) and Fe_3_O_4_/P-N-PAA, IR heated at 600 (**2b**) and 800 °C (**1c**) and MNP/P-N-PAA (**2c**).

**Figure 7 polymers-14-02935-f007:**
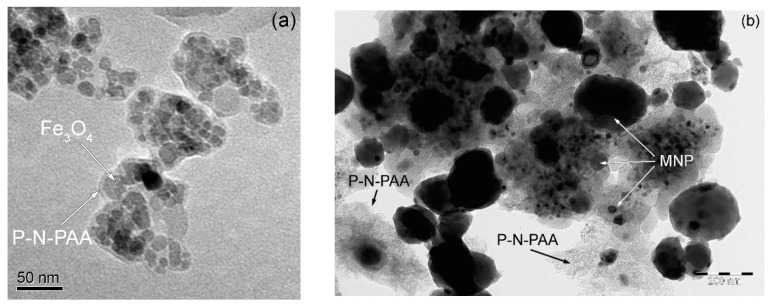
TEM images of Fe_3_O_4_/P-N-PAA (**a**) and MNP/P-N-PAA (**b**).

**Figure 8 polymers-14-02935-f008:**
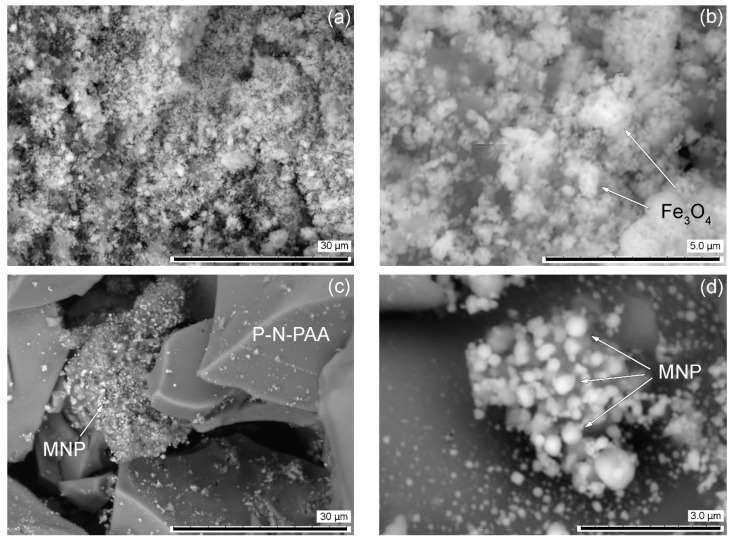
SEM images of Fe_3_O_4_/P-N-PAA (**a**,**b**) and MNP/P-N-PAA (**c**,**d**).

**Figure 9 polymers-14-02935-f009:**
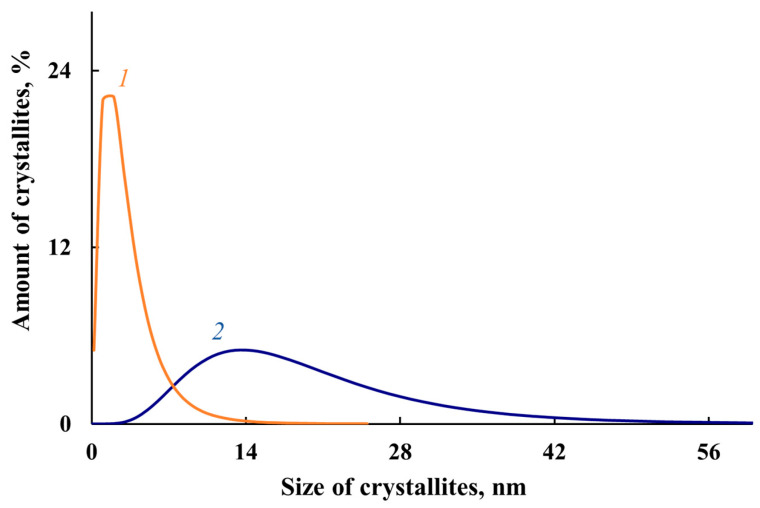
Crystallites size distribution in Fe_3_O_4_/P-N-PAA (*1*) and MNP/P-N-PAA (*2*).

**Figure 10 polymers-14-02935-f010:**
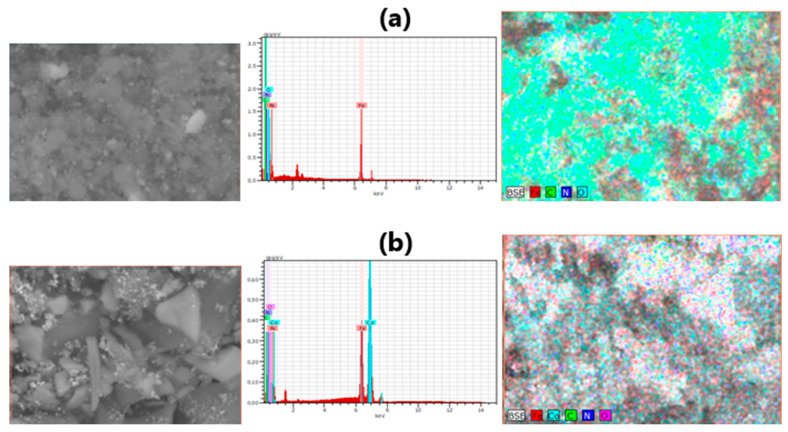
Representative SEM and EDX images of Fe_3_O_4_/P-N-PAA (**a**) and MNP/P-N-PAA (**b**).

**Figure 11 polymers-14-02935-f011:**
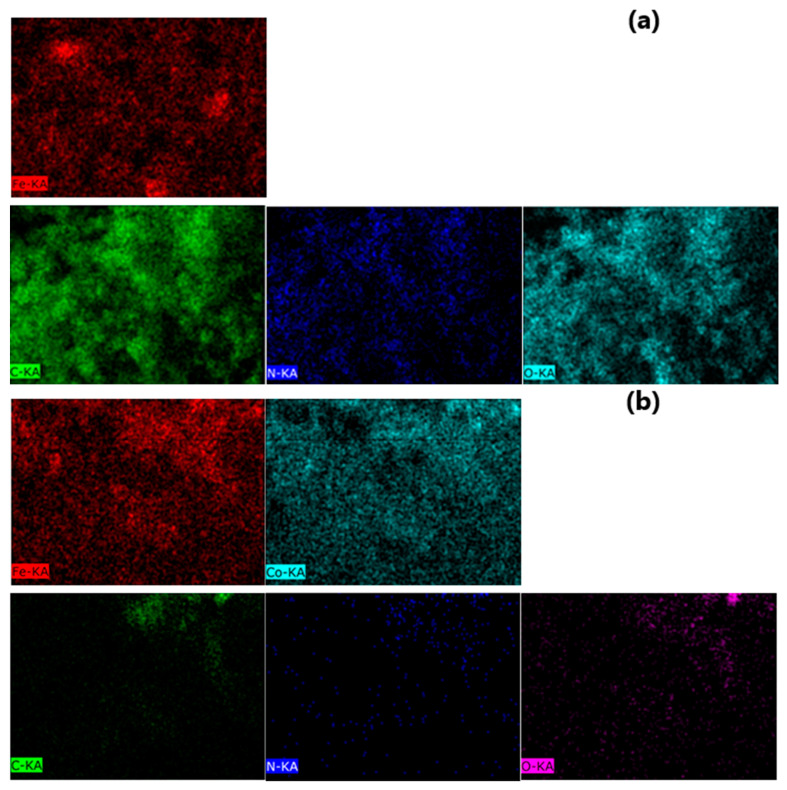
SEM-EDS mapping images of ferrum Fe, cobalt Co, carbon C, nitrogen N and oxygen O in Fe_3_O_4_/P-N-PAA (**a**) and MNP/P-N-PAA (**b**).

**Figure 12 polymers-14-02935-f012:**
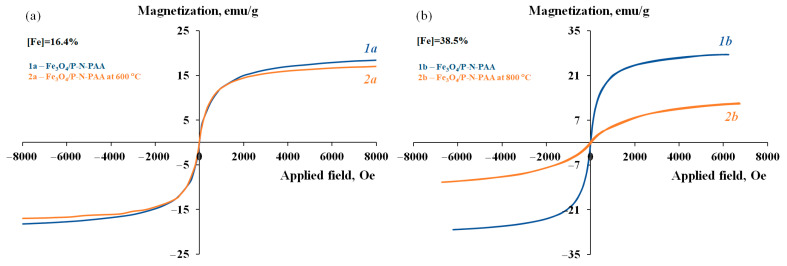
Magnetization of Fe_3_O_4_/P-N-PAA (**1a**,**1b**) and Fe_3_O_4_/P-N-PAA, IR heated at 600 °C (**2a**) and 800 °C (**2b**), as a function of applied magnetic field.

**Figure 13 polymers-14-02935-f013:**
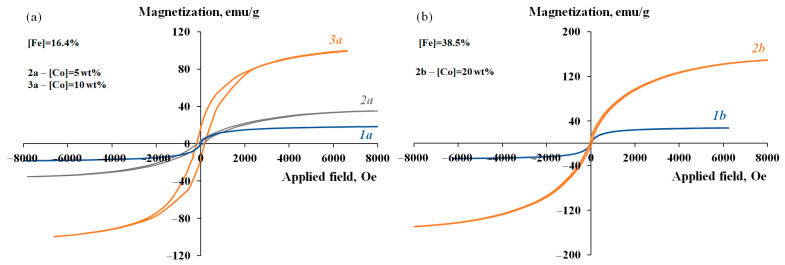
Magnetization of Fe_3_O_4_/P-N-PAA (**1a**,**1b**) and MNP/P-N-PAA, obtained at 800 °C, [Co] = 5 (**2a**), 10 (**3a**) and 20 wt% (**2b**), as a function of applied magnetic field.

**Figure 14 polymers-14-02935-f014:**
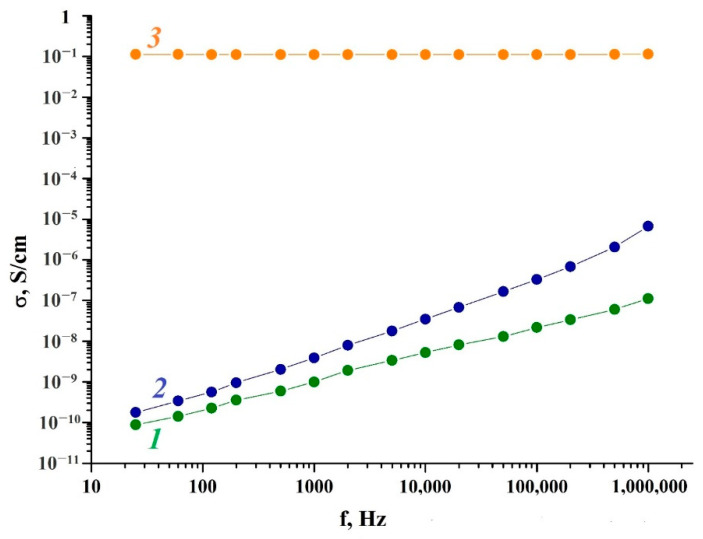
Dependence of conductivity on the frequency for P-N-PAA *(1)*, Fe_3_O_4_/P-N-PAA *(2)* and MNP/P-N-PAA, obtained at 800 °C *(3)*.

**Figure 15 polymers-14-02935-f015:**
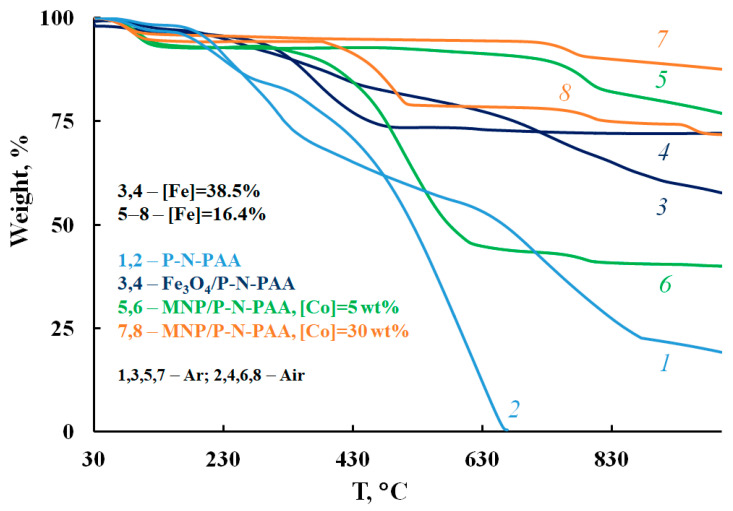
TGA thermograms of P-N-PAA *(1, 2)*, Fe_3_O_4_/P-N-PAA *(3, 4)* and MNP/P-N-PAA, obtained at 800 °C, [Co] = 5 *(5, 6)* and 30 wt% *(7, 8)*, in the argon flow *(1, 3, 5, 7)* and in air *(2, 4, 6, 8)*.

**Figure 16 polymers-14-02935-f016:**
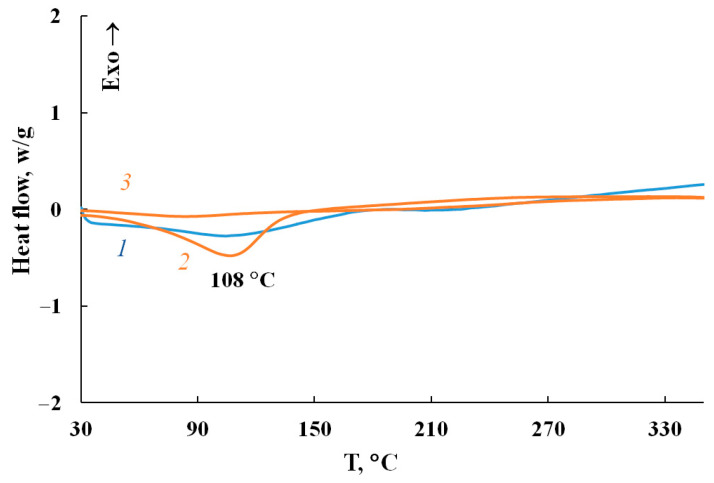
DSC thermograms of Fe_3_O_4_/P-N-PAA (*1*) and MNP/P-N-PAA (*2*, *3*) at heating in the nitrogen flow (*1*, *2*—first heating, *3*—second heating).

**Figure 17 polymers-14-02935-f017:**
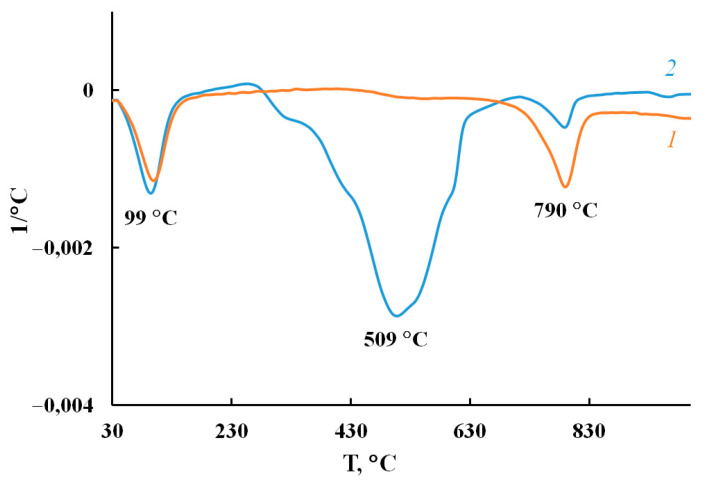
DTG curves of MNP/P-N-PAA, obtained at [Co] = 5 wt%, in the argon flow (*1*) and in air (*2*).

**Table 1 polymers-14-02935-t001:** Magnetic properties of nanomaterials.

Nanomaterials	T, °C	Co, wt%	Fe, wt%	Nano-Particles Size, nm	Me Phase Composition	*H_C_*, Oe	*M_S_*, emu/g	*M_R_*, emu/g	*M_R_*/*M_S_*
Co/PDPA [61]	450	10	-	2–8	α-Co, β-Co	145	22.23	0.69	0.03
Co/PPOA [62]	500	10	-	4–14	β-Co	134	26.33	3.05	0.116
Co-Fe/PDPA [65]	600	5	10	8–30, 400–800	Co-Fe	5	20.43	0.06	0.003
Co-Fe/PPOA [63]	600	5	10	4–24, 400–1400	Co-Fe	55	27.28	0.7	0.025

**Table 2 polymers-14-02935-t002:** Inductively coupled plasma atomic emission spectroscopy (ICP-AES) data of nanocomposites and elemental analysis data of P-N-PAA component.

Materials	[Co] *, wt%	Co, %	Fe, %	C, %	N, %	H, %	O, %	C/N	C/H
P-N-PAA	-	-	-	60.7	8.2	5.8	25.3	7.7	10.5
Fe_3_O_4_/P-N-PAA	-	-	38.5	23.58	2.75	1.88	33.3	8.6	12.5
MNP/P-N-PAA	20	29.1	58.3	8.48	0.94	0.12	3.1	9.0	70.7

* [Co] wt% at the loading.

**Table 3 polymers-14-02935-t003:** Magnetic properties of nanomaterials.

Nanomaterials	T, °C	[Co] *, wt%	[Fe] **, %	Co ***, %	Fe ***, %	**** MNP Phase Composition	*H_C_*, Oe	*M_S_*, emu/g	*M_R_*, emu/g	*M_R_*/*M_S_*
Fe_3_O_4_/P-N-PAA	0 0	- -	16.4 38.5	- -	16.4 38.5	Fe_3_O_4_ Fe_3_O_4_	0 0	18.41 27.58	0 0	0 0
Fe_3_O_4_/P-N-PAA	600 800	- -	16.4 38.5	- -	16.9 47.2	Fe_3_O_4_ Fe_3_O_4_, FeO, α-Fe, γ-Fe, Fe_4_N	0 25	17.02 12.41	0 0.28	0 0.022
MNP/P-N-PAA	800 800 800 700 800	5 10 20 20 30	16.4 16.4 16.4 38.5 16.4	8.6 13.6 28.3 29.1 38.0	19.2 17.4 14.2 58.3 17.7	Co-Fe, γ-Fe, Fe_3_C Co-Fe, γ-Fe, β-Co Co-Fe, β-Co Co-Fe, β-Co Co-Fe, β-Co	128 176 170 45 200	35.22 99.86 81.58 149.67 95.70	3.00 16.00 12.80 6.0 20.00	0.085 0.160 0.156 0.040 0.209

* [Co] wt% at the loading. ** [Fe] content in the neat Fe_3_O_4_/P-N-PAA. *** According to ICP-AES data. **** MNP—Co-Fe, γ-Fe, β-Co and Fe_3_C magnetic nanoparticles. H_C_—coercive force, M_S_—saturation magnetization and M_R_—residual magnetization.

**Table 4 polymers-14-02935-t004:** EDS analysis data of nanomaterials.

Nanomaterials	MNP Phase Composition	* Polymer Component, %	Co		Fe		C		N		O	
			wt%	at%	wt%	at%	wt%	at%	wt%	at%	wt%	at%
Fe_3_O_4_/P-N-PAA	Fe_3_O_4_	28	-	-	12.99	3.43	49.45	60.76	8.84	9.32	28.72	26.49
MNP/P-N-PAA	Co-Fe, β-Co	16	72.44	55.26	19.63	15.80	7.05	26.39	0.19	0.62	0.68	1.92

* According to TGA data.

**Table 5 polymers-14-02935-t005:** The conductivity values of materials.

Materials	* σ_*ac*_, S/cm		σ_*dc*_, S/cm	n	*A*
P-N-PAA	8.8 × 10^−11^	1.1 × 10^−7^	2.8 × 10^−12^	0.75	8.5 × 10^−12^
Fe_3_O_4_/P-N-PAA	1.8 × 10^−10^	6.7 × 10^−6^	1.3 × 10^−10^	0.99	1.3 × 10^−12^
MNP/P-N-PAA	1.1 × 10^−1^	1.2 × 10^−1^	1.1 × 10^−1^	0.98	6.5 × 10^−10^

* σ—The *ac* conductivity at 25 Hz and 1 MHz.

**Table 6 polymers-14-02935-t006:** Thermal properties of nanomaterials.

Materials	[Co] *, wt%	[Fe] **, %	Co ***, %	Fe ***, %	**** MNP Phase Composition	^ *T*_5%_, °C	^* *T*_20%_, °C	^^^^ *T*_50%_, °C	^^^^^ Residue, %
P-N-PAA	-	-	-	-	-	185/205	357/299	523/663	0/20
Fe_3_O_4_/P-N-PAA	-	38.5	-	38.5	Fe_3_O_4_	258/230	405/557	>1000/>1000	72/58
MNP/P-N-PAA	5	16.4	8.6	19.2	Co-Fe, γ-Fe, Fe_3_C	102/111	459/>1000	579/>1000	40/77
	30	16.4	38.0	17.7	Co-Fe, β-Co	108/371	507/>1000	>1000/>1000	71/88

* [Co] wt% at the loading. ** [Fe] content in the neat Fe_3_O_4_/P-N-PAA. *** According to ICP-AES data. **** MNP—Co-Fe, γ-Fe, β-Co, and Fe_3_C magnetic nanoparticles. ^ T_5%_, ^* T_20%_, ^^ T_50%_—5, 20 and 50 % weight losses (air/argon). ^^^ residue at 1000 °C (air/argon).

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
