# Peer review of "Novel Hybrid Nanomaterials Based on Poly-N-Phenylanthranilic Acid and Magnetic Nanoparticles with Enhanced Saturation Magnetization"

_polymers, 2022, doi:10.3390/polym14142935_

Round 1
Reviewer 1 Report
The manuscript has been definitely improved, yet a critical issue must be still addressed. As the authors themselves write, Raman Spectra of heat treated samples are typical for graphitic species and they do not show any residual polymer band. This could decompose leaving a high amount of carbon phase. In this view the term nanocomposite cannot be used. The authors should clarify this aspect. Solid state NMR or EPR spectroscopy could be useful to this purpose.
Author Response
The authors are grateful to the reviewer for constructive and valuable comments on the manuscript. Please find below our answers to the comments.
Comments and Suggestions for Authors
- The manuscript has been definitely improved, yet a critical issue must be still addressed. As the authors themselves write, Raman Spectra of heat treated samples are typical for graphitic species and they do not show any residual polymer band. This could decompose leaving a high amount of carbon phase. In this view the term nanocomposite cannot be used. The authors should clarify this aspect. Solid state NMR or EPR spectroscopy could be useful to this purpose.
In this work, the thermal transformations of a metal-polymer Fe3O4/P-N-PAA nanocomposite with a core-shell structure in the presence of Co (II) under IR heating have been studied. IR heating of precursors prepared by dissolving a Co (II) salt in a magnetic fluid based on Fe3O4/P-N-PAA leads to the formation of novel hybrid nanomaterials based on poly-N-phenylanthranilic acid (P-N-PAA) and cobalt- and iron-containing magnetic nanoparticles (MNP) with enhanced saturation magnetization.
A nanocomposite is a multi-phase solid material in which at least one of the phases has one, two or three dimensions less than 100 nm in size. In this case, it is important to have a clear phase boundary. The proposed MNP/P-N-PAA nanocomposites completely meet this definition, since MNP/P-N-PAA contains metal nanoparticles with sizes not exceeding 100 nm, dispersed in an amorphous matrix.
Unfortunately, solid-state CP/MAS 13C NMR spectroscopy cannot be used to study such high-magnetic-moment materials due to the strong magnetization of nanocomposites in a magnetic field. Due to short relaxation times, the presence of magnetic Co-Fe and β-Co nanoparticles causes the appearance a single broad line in the 13C NMR spectrum.
As for the use of electron paramagnetic resonance (EPR) spectroscopy, it is not possible to record the EPR spectrum of P-N-PAA because the formation of intramolecular hydrogen bonds breaks the coplanarity of the polyconjugated system, which leads to very short regions of effective conjugation with a low degree of delocalization of π-electrons. That is, it is impossible to show the presence of a polymer in the structure of MNP/P-N-PAA nanocomposite by EPR spectroscopy.
The following experimental results favor the presence of a polymer component in the MNP/P-N-PAA nanocomposite structure:
- Elemental analysis data reveal the presence of nitrogen and hydrogen atoms in the nanocomposite structure. They should be absent in the case of carbon phase formation only.
- In TGA thermograms (Fig. 14), the same shape of weight loss curves 4 and 6 show the thermooxidative degradation of polymer component in MNP/P-N-PAA (6) and Fe3O4/P-N-PAA (4). The DTG curve of MNP/P-N-PAA shows thermal decomposition of polymer component within the range of 320–620 °C, with the maximum at 509 °C (Fig. 16).
Thus, the presence of nitrogen and hydrogen atoms in the MNP/P-N-PAA structure (Table 2), as well as the absence of a sharp peak at 2θ = 39.36°, which characterizes carbon, on the XRD patterns of MNP/P-N-PAA (Fig. 4) indicate an incomplete transformation of the polymer component to all-carbon structures. During the reduction of Fe3O4 in the course of Fe3O4/P-N-PAA thermal transformations at high temperatures (700–800 °C), a Fe---OOC coordination bond formed via binding of the carboxylate-ion with iron in the core-shell structure is broken. As a result, cobalt- and iron-containing MNP of various compositions dispersed in a polymer matrix are formed. Apparently, these processes lead to the partial formation of graphite-like structures in MNP/P-N-PAA.

Reviewer 2 Report
In this manuscript, the authors prepared MNP/P-N-PAA nanocomposites under IR heating and investigated their structural, morphological, and magnetic properties. I suggest the acceptance of the manuscript after the following revision.
1. The authors claimed that the as-synthesized nanocomposites have a core/shell structure. But the TEM images show large aggregation of MNP. Can the authors point out in the TEM image where the polymer shell is?
2. I suggest the authors to conduct SEM-EDS mapping of the as-synthesized structures to characterize the metal distribution within the nanocomposites.
3. I suggest the authors to add more discussions to the XRD figures and label the main peaks.
4. Figure 3a, what's the difference between 1 and 2?
5. I suggest the authors to add a scheme of reaction between polymer and metal precursor to explain the synthesis of the nanocomposites.
6. Some papers about the application of nanocomposite magnetic materials can be cited: "Hybrid conjugated polymer/magnetic nanoparticle composite nanofibers through cooperative non-covalent interactions." Nanoscale Advances 2.6 (2020): 2462-2470.; "Effects of magnetic nanoparticles and external magnetostatic field on the bulk heterojunction polymer solar cells." Scientific reports 5.1 (2015): 1-9.; Controlled drug release and hydrolysis mechanism of polymer–magnetic nanoparticle composite." ACS Applied Materials & Interfaces 7.18 (2015): 9410-9419
7. Please check typos and grammar error over the entire manuscript.
Author Response
The authors are grateful to the reviewer for constructive and valuable comments on the manuscript. Please find below our answers to the comments.
Comments and Suggestions for Authors
- The authors claimed that the as-synthesized nanocomposites have a core/shell structure. But the TEM images show large aggregation of MNP. Can the authors point out in the TEM image where the polymer shell is?
Only the neat Fe3O4/P-N-PAA nanocomposite has a core-shell structure, where Fe3O4 nanoparticles form the core, and P-N-PAA is the shell, which makes it possible to use Fe3O4/P-N-PAA nanocomposites to obtain magnetic fluids suitable for dissolving metal salts.
During the synthesis of MNP/P-N-PAA from Fe3O4/P-N-PAA, cobalt interacts with Fe3O4 reduction products, which leads to the formation of a mixture of Co-Fe, g-Fe, β-Co, Fe3C magnetic nanoparticles (MNP) of various compositions dispersed in a polymer matrix (according to TEM and SEM). Metals are reduced due to the hydrogen released during dehydrogenation of phenylenamine units of polymer chain. As for the core-shell structure, at 800 °C, due to the reduction of Fe3O4 a Fe---OOC coordination bond formed via binding of the carboxylate-ion with iron is broken and obtained MNP are dispersed in the polymer matrix.
- I suggest the authors to conduct SEM-EDS mapping of the as-synthesized structures to characterize the metal distribution within the nanocomposites.
SEM-EDS mapping were conducted to characterize the distribution of cobalt Co, ferrum Fe, carbon C, nitrogen N and oxygen O in the nanomaterials based on P-N-PAA.
Appropriate additions were introduced into the text in colored characters.
- I suggest the authors to add more discussions to the XRD figures and label the main peaks.
Appropriate additions were introduced into the XRD figures.
- Figure 3a, what's the difference between 1 and 2?
Figure 3a shows XRD patterns of neat Fe3O4/P-N-PAA (1a) and Fe3O4/P-N-PAA, IR heated at 600 °С (2a). IR heating of Fe3O4/P-N-PAA at 600 °C does not lead to any noticeable phase changes. XRD patterns of Fe3O4/P-N-PAA and Fe3O4/P-N-PAA, IR heated at 600 °С, demonstrate only the Fe3O4 phase.
Appropriate additions were introduced into the XRD figures.
- I suggest the authors to add a scheme of reaction between polymer and metal precursor to explain the synthesis of the nanocomposites.
In this work, we focused on the method of obtaining nanocomposites, the structure and properties of the resulting nanomaterials. The study of the mechanism of formation of nanocomposites seems to be a very important and complex task that goes beyond the scope of this study.
- Some papers about the application of nanocomposite magnetic materials can be cited: "Hybrid conjugated polymer/magnetic nanoparticle composite nanofibers through cooperative non-covalent interactions." Nanoscale Advances2.6 (2020): 2462-2470.; "Effects of magnetic nanoparticles and external magnetostatic field on the bulk heterojunction polymer solar cells." Scientific reports5.1 (2015): 1-9.; Controlled drug release and hydrolysis mechanism of polymer–magnetic nanoparticle composite." ACS Applied Materials & Interfaces 7.18 (2015): 9410-9419
Proposed papers about the application of nanocomposite magnetic materials were cited.
- Please check typos and grammar error over the entire manuscript.
A professional translator has corrected typos and mistakes.

Reviewer 3 Report
Ozkan and coworkers report an interesting article about the preparation of hybrid nanomaterials. I recommend its publication after the following changes:
-Selfcitation, 10 out of 68 references belongs to the correponding authors, which I consider to be excesively high.
Regarding the Characterization techniques: The y-axis in IR spectra and x-ray difractogram does not afford any information, should be erased in Figures 3, 4, 5 and 6
Figure 5 should be normalize to detect the vibrations from compund 2 (MNP-PN-PAA) the orange one. Otherwise, it's hardly difficult to see anything.
TEM and SEM figures does not ned to be that big.
In fig 15, there's no physicial meaning for a 1/ºC over zero. So y-axis should start in zero.
Author Response
The authors are grateful to the reviewer for constructive and valuable comments on the manuscript. Please find below our answers to the comments.
Comments and Suggestions for Authors
- Selfcitation, 10 out of 68 references belongs to the correponding authors, which I consider to be excesively high.
We have enriched the source of references to reduce the self-citation rate. The self-citations is 8 out of 72 references (11.11%) and the percentage is lower than the journal standard rate (14%).
Appropriate additions were introduced into the text in colored characters.
- Regarding the Characterization techniques: The y-axis in IR spectra and x-ray difractogram does not afford any information, should be erased in Figures 3, 4, 5 and 6.
Appropriate changes were made.
- Figure 5 should be normalize to detect the vibrations from compund 2 (MNP-PN-PAA) the orange one. Otherwise, it's hardly difficult to see anything.
Appropriate changes were made.
- TEM and SEM figures does not ned to be that big.
Appropriate changes were made.
- In fig 15, there's no physicial meaning for a 1/ºC over zero. So y-axis should start in zero.
Appropriate changes were made.

This manuscript is a resubmission of an earlier submission. The following is a list of the peer review reports and author responses from that submission.
Round 1
Reviewer 1 Report
The authors report on hybrid nanomaterials with enhanced saturation magnetization. Comments provided below:
- Please add a flowchart with the step-by-step and if possible with images of the solution to illustrate color changes to guide readers.
- How was the sample prepared for the LCR-hardware?
- For the experiment performed at 800oC, could the authors address whether the mixture of magnetic materials is an advantage (or not)? For instance, what are the consequences of having a mixture of magnetic materials in the core-shell structure?
- It is not clear why "graphite like structures" are being formed during the processing. The authors should address this point in further details.
- In line 244-245, it is mentioned the contents of Fe and Co, but they do not add up to 100%. Is the remaining polymer? If so, please clarify in the text.
- Since the authors report "a significant proportion of superparamagnetic particles", please estimate their quantity (in Fig. 10 they are not). In Fig. 10b have the authors correlated quantity of magnetic material (in vol%) to check whether Ms is scaling with it? Please add these info.
- How do the values of Equation 1 correlate with Table 3?
- Line 328: DSC does not measure mass change; TGA does. Please correct it.
Reviewer 2 Report
This study is focused on the synthesys of hybrid materials made of Poly-N-Phenylan-2 thranilic Acid and Magnetic Nanoparticles with relevant saturation magnetization. Even though the authors claim the system is novel, this is not the case, since they carried out a similar study before. Please refer to the paper:
RSC, 2021, 11, 24772-24786.
Thus the manuscript has poor degree of novelty
In addition, FTIR and TG data clearly show that the polymer undergoes revelant thermal degradation. Thus, material obtained through IR heating is essentially made of bimetallic phase and carbon residue. Due to the negligible content in the polymer component, the paper subject does match the journal aims and scopus.
Furthermore, the study contains some relevant weaknesses, in particular the authors compare the features of bimetallic nanoparticles obtained after IR heating with those of Fe3O4/P-N-PAA nanoparticles which were not submitted to the same thermal treatment. Thus, they do not look as an appropriate reference material.
For all these reasons I cannot recommend this study for publication in Polymers